# On the Challenges and Opportunities in Generative AI

**Laura Manduchi,**[*][1]  **Clara Meister,**[*][1]  **Kushagra Pandey,**[*][2]  **Robert Bamler,**[3]  **Ryan Cotterell,**[1]
**Sina Däubener,**[4]  **Sophie Fellenz,**[5]  **Asja Fischer,**[4]  **Thomas Gärtner,**[6]  **Matthias Kirchler,**[5][7]
**Marius Kloft,**[5]  **Yingzhen Li,**[8]  **Christoph Lippert,**[7][9]  **Gerard de Melo,**[7][9]  **Eric Nalisnick,**[10]
**Björn Ommer,**[11]  **Rajesh Ranganath,**[12]  **Maja Waldron,**[13]  **Karen Ullrich,**[14]
**Guy Van den Broeck,**[15]  **Julia E Vogt,**[1]  **Yixin Wang,**[16]  **Florian Wenzel,**[17]  **Frank Wood,**[18]
**Stephan Mandt,**[†][2]  **Vincent Fortuin**[†][19][20]

[1]ETH Zürich; [2]UC Irvine; [3]University of Tübingen; [4]Ruhr-University Bochum; [5]RPTU Kaiserslautern-Landau; [6]TU Wien; [7]Hasso Plattner Institute; [8]Imperial College London; [9]University of Potsdam; [10]Johns Hopkins University; [11]LMU Munich; [12]New York University; [13]University of Wisconsin-Madison; [14]Meta AI; [15]UCLA; [16]University of Michigan; [17]Mirelo AI; [18]University of British Columbia; [19]Helmholtz AI; [20]TU Munich

*Reviewed on OpenReview:* `https://openreview.net/forum?id=NeS9Kj2JwF`

## Abstract

The field of deep generative modeling has grown rapidly in the last few years. With the availability of massive amounts of training data coupled with advances in scalable unsupervised learning paradigms, recent large-scale generative models show tremendous promise in synthesizing high-resolution images and text, as well as structured data such as videos and molecules. However, we argue that current large-scale generative AI models exhibit several fundamental shortcomings that hinder their widespread adoption across domains. In this work, our objective is to identify these issues and highlight key unresolved challenges in modern generative AI paradigms that should be addressed to further enhance their capabilities, versatility, and reliability. By identifying these challenges, we aim to provide researchers with insights for exploring fruitful research directions, thus fostering the development of more robust and accessible generative AI solutions.

## 1  Introduction

The past few years have demonstrated the immense potential of large-scale generative models to create powerful AI tools capable of impacting society profoundly. Large Language Models (LLMs) (Brown et al., 2020; Chowdhery et al., 2022; OpenAI, 2023; Rae et al., 2021) and their dialogue agents, such as ChatGPT (OpenAI, 2023) and Llama 3 (Grattafiori et al., 2024) have enabled the development of highly effective text generation systems that produce coherent, contextually relevant, and user-tailored outputs across a wide range of use cases. Similarly, advancements in diffusion models (Sohl-Dickstein et al., 2015; Song et al., 2020; Ho et al., 2020) have led to groundbreaking advancements in image synthesis tasks, such as large-scale text-to-image generation (Ramesh et al., 2022; Rombach et al., 2022; Saharia et al., 2022; Esser et al., 2024). These successes show that highly effective AI systems can be built using a relatively straightforward recipe: combining simple generative modeling paradigms (Larochelle & Murray, 2011; Sohl-Dickstein et al., 2015) with successful network architectures (Vaswani et al., 2017; Dosovitskiy et al., 2020; Ronneberger et al., 2015), training on large-scale datasets, and the incorporation of preferences via human feedback (Ouyang et al., 2022; Ziegler et al., 2019). The impact of generative AI has not been limited to text and image

---

[*]Equal contribution. Correspondence to `laura.manduchi@inf.ethz.ch`
[†]Equal contribution.

generation applications. It has fueled accelerated progress across a variety of research fields and practical applications, spanning from biology (Jumper et al., 2021) to weather forecasting (Ravuri et al., 2021), code generation (Chen et al., 2021b; Li et al., 2022b), video creation (Yang et al., 2023c; Ho et al., 2022b; Singer et al., 2022; Brooks et al., 2024), audio synthesis (Borsos et al., 2023; Liu et al., 2023a), and even artistic and musical composition (Huang et al., 2023b).

> With the current advancements and excitement surrounding generative AI, a question naturally arises: Are we on the brink of an AI utopia? Are we close to developing what we might call a *perfect generative model*? For the purpose of this survey, we define such a model as a single system that (i) can approximate the joint data distribution of any modality, (ii) provides calibrated uncertainty assessments and demonstrates causal consistency, and (iii) delivers controllable outputs that satisfy stringent requirements on robustness, safety, efficiency and societal alignment. We argue in this paper that the answer is a resounding *no*; rather, the realization of such a model, one that would fundamentally transform the field of AI is still hampered by substantial theoretical, practical, and ethical challenges, and incremental advances alone are unlikely to close the gap in the near term.

Amidst the excitement and anticipation surrounding this new wave of Deep Generative Models (DGMs),[1] it is easy to overlook the new set of challenges they introduce. Unlike many of the traditional machine learning models, DGMs generate outputs in very high-dimensional spaces, which introduces several technical complexities. These include significantly increased computational demands, a need for larger datasets to accurately capture the underlying data distribution, and challenges in effectively evaluating and interpreting the generated outputs (Tong et al., 2024). And while significant progress has been made in improving interpretability and computational efficiency for traditional models (Marcinkevics & Vogt, 2020), these existing methods are frequently ill-suited for DGMs (Singh et al., 2024), at least in part because of the complex and high-dimensional nature of their outputs. Consequently, there is a pressing need for the development of a new set of techniques and tools tailored to these models, particularly to enable efficient inference, interpretability and quantization. These challenges lead us to conclude that scaling up current paradigms is *not* in isolation the ultimate path towards a perfect generative model. While increasing model size and training data can enhance performance on benchmarks (Hoffmann et al., 2022), it does little to address the fundamental shortcomings of DGMs, such as their inefficiency, lack of inclusivity, limited transparency, and barriers to usability—particularly in high-stakes domains where reliability and fairness are paramount.

This work offers a collection of views and opinions from different communities about these key unresolved challenges in generative AI, with the ultimate goal of guiding future research toward what we perceive are the most critical and promising areas. Concretely, we discuss key challenges in (a) broadening the *scope and adaptability* of DGMs, i.e., their ability to robustly generalize across different domains and modalities (Section 2); (b) improving their *efficiency and resource utilization*, i.e., to lower the memory and computational requirements and enhance accessibility and sustainability in their adoption (Section 3); and, finally, (c) addressing *ethical and societal concerns* that are crucial for responsible deployment (Section 4).

By presenting a broader roadmap of the current state and open challenges in generative AI, this paper offers an integrated entry point for researchers and practitioners. Although many existing surveys offer in-depth reviews of specific subfields, such as robustness, causal modeling, or modality-specific techniques, they are often narrowly focused, making it difficult to grasp the collective landscape. In contrast, we put forward high-level insights across domains, highlight emerging research directions, and guide readers to foundational and topic-specific work. In doing so, our goal is to foster the development of generative AI systems that are more robust, inclusive, and accessible.

This paper emerged as a result of the Dagstuhl Seminar on *Challenges and Perspectives in Deep Generative Modeling*[2] held in Spring 2023.

---

[1]In this paper, we take the term Generative AI to refer to systems whose core component is a large-scale DGM. For brevity, we refer to this class and its instances simply as DGMs.

[2]https://www.dagstuhl.de/23072

## 2 Expanding Scope and Adaptability

| Sub-challenge | Typical failure mode | Promising mitigation avenues | Reference materials | Research-question candidates |
|---|---|---|---|---|
| Robust generalisation to OOD inputs (§2.1) | Large performance drop on unseen domains | Retrieval-augmented generation (RAG); Group DRO training (e.g., Sagawa* et al., 2020) | **Datasets:** WILDS; ImageNet-C **Surveys:** Shen et al. (2021); Yang et al. (2023b); Li et al. (2023c) | *Q1*: Can retrieval-augmented generators help close the WILDS gap without model retraining? *Q2*: Can we provide an inductive bias for foundation models via architectural modifications or training objectives that predisposes them to accurately capture tail events? |
| Resilience to adversarial perturbations (§2.1) | Imperceptible noise fools the model | Gaussian noise smoothing; Certified defense methods; Adversarial finetuning for diffusion | **Benchmarks:** RobustBench Leaderboard; NSFW Adversarial Benchmark **Surveys:** Sun et al. (2023b) | *Q1*: Will certifiably-robust diffusion sampling algorithms scale to large models, e.g., ones trained on ImageNet? *Q2*: How can we unify adversarial training across different modalities for multi-modal models in a single framework? |
| Mitigating learning of spurious correlations (§2.1 / 2.2) | Model predicts based on background cues, not capturing meaningful relationships | Counterfactual data augmentation; Invariant Risk Minimization-based approaches | **Datasets:** Waterbirds; Colored-MNIST **Surveys:** Ye et al. (2024) | *Q1*: How can we reliably detect and quantify hidden spurious cues encoded by foundation-model features? *Q2*: Which causal probes best expose hidden biases in high-dimensional latent spaces? |
| Capturing causal dependencies (§2.2) | Models generate statistically plausible but causally impossible outcomes | SCM-guided loss functions; Interventional training objectives | **Benchmarks:** CausalBench **Surveys:** Komanduri et al. (2024) | *Q1*: How can causal invariance objectives be integrated into generative model training to improve robustness to distribution shifts? *Q2*: Can tractable surrogate objectives approximate interventional likelihood, enabling scalable causal DGM training? |
| Accounting for implicit assumptions (§2.3) | (Implicitly) assumed characteristics of data-generating distribution do not persist under domain shift | Domain-expert knowledge integration; Statistical assumption testing (e.g., independence, stationarity checks) | — | *Q1*: How can we quantify the degree of modeling-assumption violations in a trained generative model before deploying it? *Q2*: What learning objectives are most robust when the true data-generating process lies outside the model family? |
| Cross-modal transfer in specialized domains (§2.4) | Failure/inability to link signals across modalities (e.g. ECG ↔ notes) | Contrastive multimodal pre-training (e.g., Raghu et al., 2022) | **Datasets:** MMIST-CCRCC; GMAI-MMBench **Surveys:** Shaik et al. (2024) | *Q1*: How can physiological constraints be incorporated into multi-modal pre-training objectives for medical domains? *Q2*: Which training/fine-tuning methods can achieve sufficient cross-modal alignment in low-resource clinical settings? |

Table 1: Research challenges summary table - Expanding Scope and Adaptability

State-of-the-art leaderboard rankings show the remarkable progress in model performance that has been made by scaling DGMs to massive datasets and model sizes (for instance, in text and high-resolution image synthesis). However, automatic evaluations on popular benchmark datasets cannot be our only measure of model success (Bender et al., 2021); such evaluations often fail to capture the nuanced limitations of DGMs, such as potential biases, inabilities to generalize to inputs from underrepresented or specialized distributions, and difficulties with aligning outputs with specific domain requirements. Understanding these inherent and often hidden constraints is essential for ensuring that DGMs can be reliably applied to various real-world tasks, where data characteristics, domain-specific constraints, and measures of success may differ significantly from those in standardized benchmarks (Durall et al., 2020; Daunhawer et al., 2022; Xu et al., 2024). This section analyzes some of these challenges in the context of large-scale DGMs from the lens

of their generalization capabilities (Section 2.1) and the lack of transparency in their underlying modeling assumptions (Section 2.3). We examine these fundamental challenges and provide research directions that could broaden the adaptability of DGMs to promote long-term progress in the field. We also discuss two promising avenues that have the potential to greatly enhance the scope of generative models: (i) integrating causal learning (Section 2.2) and (ii) the development of a versatile, generalist agent capable of handling heterogeneous data types (Section 2.4).

## 2.1 Generalization and Robustness

To ensure reliability across various domains, DGMs must generalize effectively under shifts to the data-generating distribution of inputs, often referred to as *out-of-distribution (OOD) robustness*, and be resilient to minor variations in the input, a necessary component of the broader notion of *adversarial robustness*. Without proper generalization, generative models may produce unrealistic or biased outputs,[3] limiting their practical utility and trustworthiness in real-world applications.

While large-scale generative models show some promise in achieving OOD robustness (Wang et al., 2023a), these models still face challenges in accurately capturing rare events or responding to adversarial inputs (Zhu et al., 2024), a difficulty that lies in effectively modeling the *tail* of information (Kandpal et al., 2023), i.e., the information that appears rarely or only once in the dataset used to (pre)train the model. This limitation indicates a gap in their ability to fully represent the vast and diverse spectrum of real-world scenarios, especially those that are less common but equally significant. Retrieval-augmented language models represent a promising approach for integrating rare or specialized knowledge into model outputs, effectively addressing challenges that cannot be resolved solely by scaling up training datasets (Kandpal et al., 2023). In the vision domain, test-time approaches, such as Generalized Diffusion Adaptation (Tsai et al., 2024), present a promising avenue towards attaining OOD robustness.

DGMs are also prone to adversarial vulnerability, often due to the presence of highly predictive but non-robust features that are used as *shortcuts* for prediction (Du et al., 2023a; Puli et al., 2023; Webson & Pavlick, 2022). This behavior poses a significant threat to various downstream scenarios, especially those of safety-critical applications (Poursaeed et al., 2021; Wang et al., 2023a). Several approaches to mitigate the effect of shortcut learning are based on model refinement or on dataset refinement, also known as data-centric approaches (Whang et al., 2021; Zha et al., 2025). In the former, work has been done towards improving robustness via adversarial training (Zou et al., 2023b; Choi et al., 2025), feature masking during training (Asgari et al., 2022), ensembling (Clark et al., 2019), contrastive learning (Choi et al., 2022), and the direct integration of prior knowledge (Ilyas et al., 2019). The latter includes improving the quality of the data used by large-scale models during training, such as through augmentation (Zhang et al., 2018), labeling (Kutlu et al., 2020)) and inference techniques—for example, employing prompt engineering (Wallace et al., 2019) or data slicing (Chung et al., 2019).

However, in most applications, foundation models are often adapted to specific tasks and downstream datasets. Standard fine-tuning techniques often overemphasize the target task, leading to catastrophic forgetting (Thanh-Tung & Tran, 2020) and a loss in the general robustness of the upstream model (Suprem & Pu, 2022). Therefore, a significant challenge is to develop robust adaptation methods that adequately solve the target task but still maintain the beneficial robustness properties of the upstream model (e.g., robustness to distribution shifts of the target dataset) (Balaji et al., 2020; Du et al., 2023a; Han et al., 2021; Liu et al., 2020). These same issues come into play when developing smaller and more efficient models for the sake of economization of DGM inference and memory costs—which we discuss in greater detail in Section 3. In this context, it is important to develop robust distillation methods that do not sacrifice the robustness of the model (Du et al., 2023b; Zi et al., 2021). We argue that two particularly promising approaches to obtaining robust and interpretable models are embedding causal structure and explicitly encoding human priors into the training process—topics we examine in the following sections.

---

[3]Here, we use the terminology *biased outputs* to refer to systematic deviations in model outputs caused by imbalances or inaccuracies in the training data and/or modeling process. These outputs then do not accurately reflect the true underlying data distribution or are skewed in ways that perpetuate inaccuracies, stereotypes, or unfair conceptions about certain outcomes.

## 2.2 Causal Generative Models

Going beyond learning mere statistical correlations and understanding how underlying factors influence the generative process is the main objective of learning a causal structure of data (Pearl & Mackenzie, 2018). Structural Causal Models (SCMs) provide the mathematical foundation for this endeavor, representing causal relationships through directed acyclic graphs paired with structural equations that encode how variables causally influence one another. Such knowledge can be used to reason about hypothetical scenarios in the world, understand the effect of interventions, and perform counterfactuals (Pearl, 2019), thus facilitating informed decision-making. Although there have been attempts to develop methods for learning the optimal generative structure of deep latent variable models from data (He et al., 2019; Manduchi et al., 2023), current generative models often neglect the underlying causal dependencies in their generative processes, making them prone to shortcut learning and spurious associations (Gururangan et al., 2018; McCoy et al., 2019).

Causal generative models have the potential to offer distribution-shift robustness, fairness, and interpretability (Schölkopf et al., 2021; Wang & Jordan, 2021). They are either focused on causal representation learning, which discovers causally related latent variables, or controllable counterfactual generation, which, instead, focuses on learning a mapping between data and known causal variables. For a detailed review of the topic, we refer to (Komanduri et al., 2024). Current open challenges include but are not limited to, scalable and robust causal discovery from observational data (Reizinger et al., 2023; Zhou et al., 2022; Montagna et al., 2024), identifiability of DGMs under weaker forms of supervision (Ahuja et al., 2023; Locatello et al., 2020; von Kügelgen et al., 2024), lack of benchmark datasets and metrics to evaluate counterfactual quality (Monteiro et al., 2023), strong assumptions that are often violated in real-world applications (Komanduri et al., 2024), and, finally, the integration of diffusion models, a field that is currently under-explored but has tremendous growth potential (Mittal et al., 2021; Pandey et al., 2022; Sanchez & Tsaftaris, 2022; Sanchez et al., 2023). We suggest that the integration of causal principles in DGMs could pave the way for the development of more robust, interpretable, and actionable generative AI systems (Zhou et al., 2023).

## 2.3 Accounting for Implicit Assumptions

**Silent Assumptions.** Current generative models often make use of implicit assumptions and inductive biases. Many of these, such as translational equivariance in CNNs or locality in audio diffusion models, are principled and empirically validated. Others, however, persist mainly for computational convenience[4] or historical precedent, even when their validity for specific applications remains unexamined or is blatantly known to be wrong (Zhao et al., 2018). As one example, the algorithms used in machine learning often assume that data are drawn independently. In reality, data points are often correlated, such as in time-series data or through repeated measurements from the same individual (Jiang & Nguyen, 2007; Kirchler et al., 2023). As another example, most generative models assume that latent distributions can be modeled on simple topological structures. However, latent distributions typically benefit from more expressive approaches (Stimper et al., 2022), suggesting the assumptions of their simplicity may be ill-founded.

We argue that convenience should not be the driving factor behind modeling assumptions. While the impact of model misspecifications on downstream applications in DGMs are not yet well understood, we have preliminary evidence suggesting their effects are undesirable: In traditional statistical analyses, such misspecifications are observed to have immense impacts (Cardon & Palmer, 2003); more recently, empirical studies have revealed systematic biases in DGMs that may stem from inadequate modeling assumptions (Zhao et al., 2018), and models that rely heavily on the training data distribution have been observed to exhibit bias and decreased performance if not properly corrected by meaningful modeling assumptions (Fortuin, 2022).

There has been some progress towards developing methods that allow practitioners to encode more precise and complex modeling assumptions. As concrete examples, random effects (Jiang & Nguyen, 2007)—the paradigm used by traditional statistical methods to model data dependencies—have been adapted to work with neural models (Simchoni & Rosset, 2023). In normalizing flows, data dependencies can be incorporated

---

[4]By "convenience" we mean design choices adopted because they are easy to implement or tractable to optimize, not because they have been shown to match the true structure of the data.

directly into the likelihood objective (Kirchler et al., 2023), an approach that might be extended to other probabilistic approaches such as VAEs and diffusion models (Sutter et al., 2023). Causal models can also be integrated to directly model data dependencies and perform counterfactual inference (Pawlowski et al., 2020)—which we discuss in more detail in Section 2.2. Notably, these methods have yet to achieve widespread adoption, despite addressing issues that are prevalent and influential in many applications. We believe that further research into the effects of implicit modeling assumptions and methods that allow a wider range of modeling assumptions are promising and impactful directions for the field.

**Incorporation of Prior Knowledge.** Recent breakthroughs in DGMs have primarily been achieved in settings where models could be trained on internet-scale data (OpenAI, 2023; Rombach et al., 2022). However, many real-world applications, such as drug design (Vamathevan et al., 2019), material engineering (Wei et al., 2019), personalized medicine (MacEachern & Forkert, 2021), and protein biochemistry (Bonetta & Valentino, 2020), often have much smaller datasets due to the high cost of data generation. In these areas, domain experts often possess extensive prior knowledge, which could potentially be leveraged to enable more data-efficient learning in generative AI models. Indeed, it has been shown in the context of VAEs that incorporating domain prior knowledge can significantly improve model performance (Fortuin et al., 2020; Jazbec et al., 2021) and even unlock their use for tasks that were previously impossible (Fortuin et al., 2019; Manduchi et al., 2021; 2022).

There are multiple routes via which prior knowledge can be encoded in generative AI systems (Dash et al., 2022). One straightforward way to incorporate domain knowledge is in Bayesian settings through the choice of prior distribution; such distributions can explicitly encode known properties of the target data. For example, an informed prior can reflect physiological constraints in medicine or chemical properties in materials science (Sam et al., 2024), taking a step towards ensuring that the learned model aligns with real-world principles. Similarly, recent work in diffusion models employs heavy-tailed priors to model extreme or rare events (Pandey et al., 2025). Therefore, we need generative models that can learn mappings between domain-specific priors and the observed data distributions more flexibly (Albergo et al., 2023; Lipman et al., 2023). Beyond priors, domain knowledge can guide architectural design by suggesting specialized network components or hierarchical structures that reflect known relationships within the data (Andreas et al., 2016b; Shen et al., 2019; Bronstein et al., 2021) or can encourage models to process data in a more human-like manner for the sake of interpretability (Andreas et al., 2016a; McCoy et al., 2020; Vu et al., 2023; Lu et al., 2023). Finally, constraints embedded in either the model specification or the training algorithm can further ground generative models in real-world processes, leading to improved performance and trustworthiness (Raissi et al., 2019; Ren et al., 2020; Dash et al., 2021; Mohan et al., 2023). Each of these approaches can equip our models with helpful inductive biases that aid data-efficient learning.

While designing future models with domain-informed inductive biases holds great promise, it also presents several challenges that must be carefully considered (Battaglia et al., 2018; Bronstein et al., 2021). For example, while VAEs are Bayesian models and, therefore, offer a natural paradigm for specifying a prior distribution over their latent space, many other DGMs lack such explicit mechanisms for encoding prior information. We consider diffusion models as a concrete example. At first, it might seem that the diffusion process's Gaussian sampling distribution is comparable to the Gaussian latent prior in a VAE, suggesting a straightforward route for specifying priors for these models. However, this property of the diffusion process arises from the central limit theorem rather than from precise knowledge about the nature of the underlying data-generating distribution. Recent works have attempted to enhance the space of diffusion priors through auxiliary dimensions (Pandey & Mandt, 2023; Singhal et al., 2023), and through alternative structured or learned priors (Trippe et al., 2023; Wu et al., 2024). Unfortunately, those approaches do not offer the same degree of flexibility of prior specification provided by the Bayesian priors in VAEs' latent space, highlighting the need for continued research into priors for diffusion models. More broadly speaking, by definition, biases constrain or push our models towards certain solutions (Mitchell, 1980). If the underlying bias does not capture every facet of the real-world process—an especially common concern in areas like biology, where core mechanisms remain poorly understood—it may inadvertently limit the model's expressivity or lead to systematic errors. That is, models may fail to learn important patterns that fall outside the imposed structure (Ghassemi et al., 2020). Adding constraints also often introduces computational challenges: physically or biologically inspired restrictions might be non-differentiable or otherwise difficult to incorporate into standard

training pipelines, leading to more complex optimization procedures or increasing computational overhead, for example when enforcing PDE/ODE constraints or discrete combinatorial rules (Cranmer et al., 2020; Li et al., 2023e). Thus, while biasing models with domain knowledge can significantly improve data efficiency and performance, careful consideration of both the correctness of those biases and the technical feasibility of their implementation is essential.

### 2.4 Foundation Models for Domain-specific and Heterogeneous Modalities

While there has been tremendous progress in large-scale foundation models for modalities like text and images, as the scope of their application widens to encompass a broader range of data modalities, a variety of challenges surrounding cross-modal alignment, data interoperability, privacy, and evaluation emerge. These challenges are particularly pronounced in specialized fields such as healthcare and chemistry (Raghupathi & Raghupathi, 2014; Korshunova et al., 2022; Busch et al., 2025).

In healthcare, generation based on diverse data types—including imaging, health records, and genomics— poses challenges in interoperability, data privacy, and security (Moor et al., 2023a). Time series generation, in particular, requires addressing irregularly sampled data, missing values, seasonality, and long-term dependencies (Steinberg et al., 2021). In chemistry, physics, and chemical engineering, generative models have huge potential, not just for molecule, drug, and material design, but also in data augmentation, property prediction, and reaction prediction (Winter et al., 2019; Ahmad et al., 2022; Castro Nascimento & Pimentel, 2023; Hu et al., 2020). Data in these fields are often sparse, heterogeneous and correlated. On the other hand, they provide a vast body of physical and chemical domain knowledge, ranging from (strict) laws of nature and boundary conditions to (soft) empirical correlations and human experience. Therefore, developing hybrid (ML + domain knowledge) foundation models is a particular challenge. While there has been some recent progress toward this goal, e.g., in the realms of physics (Jirasek et al., 2022; Jirasek & Hasse, 2023; Howard et al., 2022) and medicine (Raghu et al., 2022; Moor et al., 2023b; Xia et al., 2024), there is still much work to be done (Venkatasubramanian, 2019).

We argue that an overarching goal of the generative modeling field is to build general models that can seamlessly integrate information from diverse sources and understand complex relationships across different types of data (Li et al., 2023a; Reed et al., 2022; Driess et al., 2023). This multi-modal integration challenge becomes particularly evident in embodied AI applications, where generative models must serve multiple functions simultaneously. We can thus view *embodied agents* as a natural testing ground for truly general generative AI because of their broad requirements: (1) generative world modeling to predict future states and simulate outcomes of potential actions, (2) multi-modal generation to produce coherent plans that span language instructions, visual predictions, and motor commands, and (3) conditional generation that respects physical constraints while taking into account dynamics of the environment. Unlike purely digital applications, where errors may be aesthetic or semantic, embodied systems expose fundamental limitations in our generative models through physical failure, e.g., a robot failing to grasp objects or navigate spaces. However, developing these systems faces a significant hurdle: data scarcity. While datasets for natural language or images are relatively accessible (Hausknecht et al., 2020; Li et al., 2023b; 2024f), multi-task, multi-environment datasets of control trajectories (states, actions, and outcomes) remain comparatively rare, hindering progress on the development of generalist embodied agents. Generative simulation is one route we identify to achieve this potential use case for cross-domain generative models (Xian et al., 2023; Fan et al., 2022).

## 3 Optimizing Efficiency and Resource Utilization

Efforts to scale DGMs for tasks like language modeling and text-to-image synthesis often involve training large models with billions of parameters, which demands significant computational resources. This leads to practical issues such as high energy costs (Wu et al., 2022) and expensive inference, limiting access for many users. This further raises environmental concerns due to the energy consumption required to fuel modern tensor processing hardware (Strubell et al., 2020). Training PaLM leads to 271 tons of CO2e effective emissions (Chowdhery et al., 2022) and training GPT-3 emits roughly twice as much under comparable accounting assumptions (Patterson et al., 2022). Therefore, there is a clear need to reduce the memory and

| Sub-challenge | Typical failure mode | Promising mitigation avenues | Reference materials | Research-question candidates |
|---|---|---|---|---|
| Efficient attention mechanisms (§3.1) | Long contexts needed in certain settings and context length limited by computational demands | Hardware-aware attention computations (e.g., FlashAttention-2); Sub-quadratic attention alternatives | **Datasets:** LongBench **Surveys:** Tay et al. (2022a) | *Q1*: Can (non-autoregressive) selective-state SSMs (e.g. Mamba) achieve the same performance as transformers in text generation tasks? *Q2*: Can better retrieval methods in RAG systems mitigate the need for longer context windows in LLMs? |
| Low-bit quantization without quality loss (§3.1) | Sharp accuracy drop when reducing FP precision to less than 4-bits | Activation-aware weight quantization; Quantization-aware training and fine-tuning | **Surveys:** Zeng et al. (2025) | *Q1*: Do activation-aware quantization methods preserve model calibration after preference-based fine-tuning (e.g., RLHF)? *Q2*: What theoretical limits bound post-training quantization of diffusion models? |
| Fast sampling for diffusion models (§3.1) | Hundreds of network evaluations needed per sample | Progressive Distillation; Consistency Models; Model quantization | **Benchmarks:** FID/IS on CIFAR-10/ImageNet **Surveys:** Shen et al. (2025) | *Q1*: Can we design models that achieve one-step generation while maintaining diffusion models' training stability and sample quality? |
| Reliable quality metrics (§3.2) | Automatic evaluation metrics do not correlate with human perception of quality | Generative models for quality assessment (e.g., LLM-as-Judge, Auto-J); Sample-based metrics (e.g., Feature-Likelihood Score) | **Benchmarks:** JudgeBench **Surveys:** (Betzalel et al., 2024) | *Q1*: How can learned reward models be made reproducible enough to serve as public benchmarks? *Q2*: Is a unified multi-modal MAUVE variant feasible? *Q3*: How reliable are LLM-as-Judge scores in OOD settings? |
| Compute-efficient model selection (§3.2) | Brute force grid-search approaches to model selection are prohibitive for today's large models | Scaling-law extrapolation; Zero-cost proxies | **Benchmarks:** NAS-Bench-101; NATS-Bench **Surveys:** Li et al. (2024a) White et al. (2022) | *Q1*: Can information-theoretic complexity measures be used during model architecture search to reliably rule out entire classes of models? *Q2*: Can zero-cost proxies and scaling law extrapolations be effectively combined to provide a stronger indication of optimal models than their individual signals? |

Table 2: Research challenges summary table - Optimizing Efficiency and Resource Utilization

computational requirements of large-scale DGMs to enhance accessibility and sustainability (Bender et al., 2021).

In this context, we discuss the efficiency-related challenges in current DGMs. We focus on minimizing training and inference costs (3.1), as well as highlighting challenges in designing evaluation metrics for DGMs (3.2), which greatly affect the computational resources needed for model selection and tuning.

## 3.1 Efficient Training and Inference

**Network Architecture.** Optimizing the network architecture, which forms the backbone of modern machine learning, is crucial for efficient training and inference in DGMs. While we have seen recent improvements in model quality (OpenAI, 2023; Touvron et al., 2023; Peebles & Xie, 2023), there is still a dearth of systematic comparative studies of architectural components' contributions to generative model performance. For instance, several popular LLMs like PaLM (Chowdhery et al., 2022) and Llama (Touvron et al., 2023) still largely reuse the original transformer architecture from Vaswani et al. (2017) with some additional modifications (Shazeer, 2020; Su et al., 2024; Zhang & Sennrich, 2019). A modification of particular importance has been that of the self-attention (Bahdanau et al., 2015) mechanism; in the original architecture, this operation incurred a computational cost that scaled quadratically in the context length. This made inference computationally expensive, especially for long-context modeling. Several recent works have proposed attention variants that provide faster inference times (Tay et al., 2022b). For example, Flash Attention (Dao

et al., 2022) employs hardware optimizations and efficient memory management techniques to reduce the effective computational overhead of attention from quadratic to linear in the context length; Flash Attention 2 (Dao, 2024) takes these optimizations a step further, bringing attention computations close to the achievable bounds on fast matrix multiplication. Grouped Query Attention (Ainslie et al., 2023) proposes a structural change to the standard attention mechanism, where queries[5] are divided into distinct groups that are then processed independently and simultaneously. We see several promising research directions for reducing the computational needs of large-scale generative models, including specialized methods that make popular network architectures more computationally efficient (e.g., the attention variants discussed above), early-exit designs that allow models to make predictions without running the forward pass through the full network (Chen et al., 2024b) and alternative autoregressive sequence-modeling frameworks with favorable properties like scalability and linear complexity in the context length (Gu & Dao, 2023; Gu et al., 2021).

Similarly, several popular large-scale text-to-image diffusion models like DALL-E 2 (Ramesh et al., 2022) and Stable Diffusion (Rombach et al., 2022) largely reuse the popular UNet (Ronneberger et al., 2015) backbone from Ho et al. (2020), which has high memory costs. Therefore, we believe that a principled study of the impact of different network components in large-scale generative models is crucial for efficient training and inference. Some recent works (Hoogeboom et al., 2023; Karras et al., 2023; Peebles & Xie, 2023; Podell et al., 2024) already explore architectural design choices for reducing diffusion model sizes, thereby improving training dynamics while enabling faster inference with a lower memory footprint.

**Model Quantization.** The goal of model quantization is to reduce the precision of model weights and activations, enabling faster, memory-efficient training and inference, ideally without losing performance on downstream tasks. The most common quantization approaches are Post-Training Quantization (PTQ), which applies quantization to a pre-trained large model to enable faster and memory-efficient inference, and Quantization-Aware Training (QAT), which involves training a quantized model from scratch (Krishnamoorthi, 2018).

Despite some progress in developing PTQ and QAT methods for LLMs (Dettmers et al., 2022; Liu et al., 2023b; Xiao et al., 2023; Yao et al., 2022; Dettmers et al., 2023) and large-scale text-to-image diffusion models (Li et al., 2023d), the existing methods are far from perfect. For instance, OPTQ (Frantar et al., 2023), a PTQ-based approach, can perform inference for a quantized LLM (in this case OPT (Zhang et al., 2022)) with 175B parameters on a single A100 GPU with 80GB of memory without degradation in accuracy. Though impressive, even this quantized model would likely have limited utility on a consumer-grade GPU device, let alone on standard edge devices. Similarly, QAT-based approaches can often achieve lower bitrates but trade off additional training for this efficiency. This can be a major computational bottleneck for large generative models. While some recent work suggests preliminary success in this direction (Lin et al., 2024a), we believe that investigating the impact of model quantization at low bitrates in large-scale generative models is a crucial direction for the practical deployment of these models.

**Design Challenges.** The current dominant modeling paradigms in generative AI, such as diffusion models (Ho et al., 2020) and LLMs (OpenAI, 2023), demonstrate remarkable sample quality. However, the design of the generative processes in these approaches can cause significant challenges. Diffusion models, for instance, rely on an iterative, multi-stage denoising process, which slows down inference considerably. Generating high-quality samples often requires hundreds to thousands of network function evaluations (NFEs) (Ho et al., 2020; Song et al., 2020). Similarly, LLMs employ an autoregressive structure that generates tokens sequentially, resulting in slow inference due to the left-to-right generation process. These challenges contrast with alternative generative models like VAEs and GANs, which require only a single NFE for sample generation. However, these models suffer from other drawbacks, such as blurry sample generation in VAEs (Dosovitskiy & Brox, 2016) and mode collapse in GANs (Arjovsky et al., 2017).

To address the inefficiencies in diffusion models, researchers have explored multiple complementary approaches to speed up inference. Some notable approaches include: developing training-free samplers (Song et al., 2021; Liu et al., 2022a; Lu et al., 2022; Zhang & Chen, 2023; Karras et al., 2022; Pandey et al., 2024), designing better diffusion processes (Singhal et al., 2023; Dockhorn et al., 2022; Pandey & Mandt,

---

[5]In the attention mechanism, a query is a transformed vector representation, typically derived from input tokens.

2023; Karras et al., 2022), and combining other model families with diffusion models (Pandey et al., 2022; Zheng et al., 2023b; Xiao et al., 2022; Yang & Mandt, 2023). Additionally, training a diffusion model in the latent space of a lossy transform (Vahdat et al., 2021; Rombach et al., 2022) not only improves memory requirements and sampling efficiency but also provides access to a more interpretable low-dimensional latent representation. A lossy transform (such as VQ-GAN (Esser et al., 2021)) can drastically reduce data dimensionality while retaining the perceptually relevant details of high-resolution images. Designing more efficient lossy compression operations in the context of diffusion models has received less attention in the community and is an important direction for further work (Yang et al., 2023d; Havasi et al., 2019; Yang et al., 2020). Despite these advances, sampling from diffusion models remains computationally challenging, typically requiring 25-50 NFEs to generate high-quality samples. While approaches based on progressive distillation (Salimans & Ho, 2022; Meng et al., 2023) can further speed up inference, they trade off additional training for faster sampling. Therefore, there is a need for DGMs that inherit all the advantages of diffusion models while supporting one-step sample generation by design (e.g., see consistency models (Song et al., 2023; Song & Dhariwal, 2024) for recent work in this direction).

In the case of LLMs, in addition to an expensive self-attention operation in transformer-based autoregressive models, sequential token generation in a left-to-right fashion in these models makes inference more expensive. Indeed, it is one reason that (sub)word tokenization—the pre-processing of text into pre-defined units—is still an essential part of these pipelines. Notably, tokenization itself introduces a strong inductive bias into language modeling: the model is constrained to work with the predefined units set by the tokenization scheme. The representation of the data that the model learns is inherently shaped by these units, limiting the model's flexibility. Token-free approaches have been proposed to allow for the joint optimization of text segmentation alongside other parameters, but in practice, they are often computationally infeasible with attention-based architectures because handling raw character sequences at scale magnifies the already expensive attention mechanism (Xue et al., 2022). Dynamic tokenization schemes (Ahia et al., 2024, e.g.,) present an interesting research direction, as they allow for predicting token boundaries at inference time, but they likewise can introduce significant computational overhead. Tokenization remains a key design choice that shapes both the performance and efficiency of modern generative models and whose further optimization is constrained by needs for efficiency (Rust et al., 2021; Toraman et al., 2023; Ali et al., 2024). Methods such as speculative decoding (Leviathan et al., 2022) are one approach that can help reduce the bottleneck caused by left-to-right generation. Non-autoregressive models also offer an interesting alternative for sequence modeling. For example, diffusion models amortize the computational cost of generating sequences across all tokens simultaneously (Dieleman et al., 2022; Wu et al., 2023; Li et al., 2022a). However, these models inherently lack the inductive bias for contextual generation, which has been shown to work well empirically for sequential modeling tasks. This affects their performance in downstream tasks that might require long-context modeling, such as video synthesis (Yang et al., 2023c; Ho et al., 2022a). While diffusion models can be incorporated within the autoregressive framework for such tasks, the resulting models can be very expensive during inference (due to the cost of synthesizing a single token using diffusion across multiple tokens). Therefore, we identify a potential tradeoff between long context modeling and efficient inference, with the diffusion and autoregressive modeling paradigms falling on the opposite ends of this tradeoff. Hence, designing generative modeling paradigms that can optimally balance this tradeoff remains challenging.

### 3.2 Evaluation Metrics

Evaluation metrics are crucial in guiding research, as the conclusions derived from empirical studies depend greatly on the chosen metrics. In modern ML, evaluation metrics are additionally a key component in hyperparameter tuning and model selection; their design thus affects computational resources required during large-scale training. However, designing robust and meaningful evaluation metrics for DGMs is challenging for several reasons.

**Evaluation Metric Design.** Many generative models are probabilistic, making likelihood-based metrics a seemingly natural choice for evaluating their performance. These metrics have been widely utilized in the literature due to their alignment with the probabilistic frameworks of such models. However, empirical evidence suggests that likelihood-based metrics often do not provide an accurate assessment of generation quality (Theis et al., 2016). In particular, they often fail to correlate with human judgments of sample

quality (Kolchinski et al., 2019; Pimentel et al., 2023). Moreover, many popular generative models do not even allow for tractable likelihood computation. Other automatic evaluation metrics are thus necessary for evaluating the quality of generated samples.

Several notable evaluation metrics for generative models follow the general paradigm of comparing the distribution of generated samples to that of train/test samples (Sajjadi et al., 2018; Pillutla et al., 2024; Jiralerspong et al., 2023). For instance, the Fréchet inception distance (Heusel et al., 2017), which is widely used for evaluating image synthesis models, takes this approach (Salimans et al., 2016; Bińkowski et al., 2018). However, these metrics are far from perfect. First, robust computations of these metrics require a large set of samples (around 50k for image generation models). This can be computationally demanding for generative models with a sequential inference process, like diffusion and autoregressive models. Even given large sample sizes, these metrics have still demonstrated issues with robustness. For example, FID can be sensitive to minor perturbations in the input data (Parmar et al., 2022) (see Chong & Forsyth (2020) for additional discussion on sources of bias associated with FID and Borji (2022) for more related evaluation metrics). Second, these methods typically rely on an external pretrained model, e.g., the GPT-2 family of language models (Radford et al., 2019) or a classifier network trained on ImageNet (Deng et al., 2009). This property makes the metric effective for evaluating sample quality within the domain of the pretrained model's data but seemingly causes it to overlook significant features or overemphasize arbitrary ones in other domains (Kynkäänniemi et al., 2023; Pimentel et al., 2023).

Recent works have attempted to improve upon the above-mentioned shortcomings. For example, Jayasumana et al. (2024) propose the use of embeddings from the CLIP model (Radford et al., 2021), which aligns images and text in a shared embedding space, in order to make a more robust evaluation metric for image synthesis models. Several evaluation metrics for text generation systems use LLMs to score or rank samples, either via prompting (Li et al., 2024g) or explicit fine-tuning on the task of evaluation (Li et al., 2024e). These metrics correlate remarkably well with human judgments (Kocmi & Federmann, 2023) and offer more fine-grained assessments of text generation systems—providing feedback at the individual sample level and taking into account user-specified criteria (Jiang et al., 2024a). These methods make progress towards broader applicability across domains and greater alignment with human evaluations but also demonstrate an increased reliance on generative models for the evaluation of other generative models. This circular dependency introduces the risk of amplifying existing model biases (Fang et al., 2024) and narrowing the diversity of model-generated content (Doshi & Hauser, 2024; Gambetta et al., 2024), as the underlying paradigm of such evaluation metrics should lead to the favoring of outputs that align with the characteristics and biases of the models used for assessment. Some works have proposed (either explicitly or implicitly) rewarding sample diversity in evaluation metrics (Zhu et al., 2018; Alihosseini et al., 2019; Jiralerspong et al., 2023), which would alleviate the latter problem. However, there is often a quality-diversity trade-off (Caccia et al., 2019; Zhang et al., 2021; Naeem et al., 2020), where a model that generates high-quality samples might have low diversity across its samples and vice versa. Further, the quantification of diversity is in itself a difficult task (Tevet & Berant, 2021).

**Subjective aspects in generation.** A major challenge underlying the evaluation of generation quality is the subjective nature of sample attributes, such as realism, fluency, and style. While human inspection is typically the gold standard for evaluating generative models (Denton et al., 2015; Zhou et al., 2019; Saharia et al., 2022), in many cases, human judges disagree over several attributes, such as which samples have better quality (Clark et al., 2021) or what is considered realistic in the target domain (e.g., medical images or industrial optical inspection). This challenge is even present for conditional synthesis tasks when the set of suitable outputs is limited by constraints from the input. For example, in text-to-image generation (Ramesh et al., 2021; Rombach et al., 2022; Saharia et al., 2022), human evaluators may have different opinions on how closely a generated image aligns with the provided description.

For this reason, a common approach is to collect numerous human judgments and set up benchmarks based on these collective scores, e.g., the Open Parti Prompts Leaderboard[6] for image generator evaluation or TURINGBENCH (Uchendu et al., 2021) for language generator evaluation. While such approaches— along with attempts to standardize human evaluation practices (Elangovan et al., 2024)—help make human

---

[6]https://huggingface.co/spaces/OpenGenAI/parti-prompts-leaderboard

evaluations a more reliable signal for guiding the development of generative models, there are several other issues with human evaluation. These include its monetary costs, the general inconsistency of human raters (Clark et al., 2021; Belz et al., 2023), and its focus on individual samples, overlooking how well the generative model reflects the data distribution as a whole.

**Evaluating Model Uncertainty and Calibration.** Model uncertainty and calibration have become quantities of interest because of their implications for the reliability, interpretability, and safety of generative AI systems. Here, we use the term model uncertainty to refer to the degree of confidence a model has in its outputs; model calibration refers to the degree to which a model's estimated probability of an event is consistent with that event's true probability of occurring.[7] As concrete examples of the importance of these metrics, high model uncertainty in language generation tasks has been linked with the occurrence of hallucinations—instances where the model produces outputs that are implausible or factually incorrect (van der Poel et al., 2022; Zhang et al., 2023c); autonomous driving systems increasingly use generative models to predict the trajectories of other vehicles, cyclists, and pedestrians (Yuan et al., 2020), and miscalibration of the probabilities of such trajectories can lead to severe accidents.

Historically, relatively simple metrics have been employed for measuring uncertainty and calibration in machine learning. For example, Shannon entropy (Shannon, 1951) has been a common metric for quantifying total model uncertainty (Houlsby et al., 2011; Depeweg, 2019); expected calibration error (Pakdaman Naeini et al., 2015, ECE) has often been employed for assessing model calibration (Guo et al., 2017; Dormann, 2020) (see Abdar et al. (2021) and Wang (2023) for detailed surveys on uncertainty and calibration in deep learning, respectively). While these metrics are well-suited to models for simpler classification problems, their extension to generative models is non-trivial. For example, language models operate over a countably infinite output space (i.e., the set of all possible strings), making exact computation of metrics like entropy or ECE infeasible. Consequently, a key aspect of research on these characteristics in generative models has been on defining metrics that are suited to them (Zhao et al., 2021; Ran et al., 2022; Luo et al., 2023; Zhao et al., 2023b; Fei et al., 2023).

To complicate matters further, there is debate regarding which definitions of these metrics actually provide useful insights about a generative model. Model uncertainty, for instance, can come from multiple sources, such as aleatoric uncertainty (intrinsic noise in the data) or epistemic uncertainty (uncertainty in the model parameters) (Hüllermeier & Waegeman, 2021; Wimmer et al., 2023). Depending on the specific use case for a model, one may be interested in the contribution of only one of these sources rather than in total model uncertainty (Osband et al., 2023; Giulianelli et al., 2023; Kuhn et al., 2023). With respect to calibration, it is unclear exactly which distribution a generative model should be calibrated to (Koevering & Kleinberg, 2024); often times, we are more interested in modeling the distribution of high-quality outputs than the data-generating distributions (Ouyang et al., 2022), albeit the data from the latter is what models are often trained on (Kalai & Vempala, 2023). These choices must be carefully and thoughtfully considered, as they play a critical role in shaping the development of methods to quantify these metrics or address poor model performance in terms of these metrics.

**Model Selection.** Model selection, i.e., identifying which model configuration or set of hyperparameters will perform best on a given task, is essential in training large-scale generative models. Evaluation metrics play a critical role here. Using knowledge of scaling laws (Kaplan et al., 2020; Henighan et al., 2020), evaluation metrics can be used to predict early on in a training run whether a model is likely to be successful (OpenAI, 2023). Recent work has shown that these predictions can be done quite precisely (Ruan et al., 2024), potentially reducing the need to train numerous large neural networks in the search for a single good model. Zero-shot proxies (Abdelfattah et al., 2021)—metrics computed on an untrained or minimally initialized network that approximate its final task performance before any training is done—are another promising research direction for compute-efficient model selection. We also believe that more effort should be invested in analyzing models' *performance-complexity* tradeoff, an important yet under-investigated measure for real-world applications at scale. This tradeoff refers to the balance between model performance and computational complexity. We argue that model selection and evaluation should perhaps shift towards identifying the model families that lie in the associated Pareto set that optimizes this tradeoff (Devroye, 2010; Braverman, 2005;

---

[7]We note that other definitions for the terms have been used; we employ this one as it is the most relevant for our exposition.

Chen et al., 2022; Braverman, 2023), as optimizing for these characteristics in isolation does not account for real-world constraints. The naïve approach—training well-performing models in each of the model classes under consideration and computing their respective computational complexities—is time- and resource-intensive. We posit that alternative assessments of complexity from information and learning theory (e.g., Xu et al., 2020) could provide the basis for more efficient metrics in these types of evaluations.

**Looking Forward.** The multi-faceted nature of what defines a high-quality generative model makes designing robust and meaningful evaluation metrics a particularly challenging task. Instead of relying on human priors about what constitutes a good quantitative metric of model quality, developers have increasingly turned to the strategy of learning reward functions directly from human preferences (Ouyang et al., 2022). This approach should allow for evaluation metrics that are more aligned with human judgment, as the reward functions are directly informed by human feedback rather than predefined criteria. These reward functions could serve as the foundation for new evaluation frameworks for generative models, and we hope they will be open-sourced to enable the development of publicly accessible benchmarks.

Evaluation metrics can help us understand and ultimately mitigate model shortcomings. While this approach has been embraced for improving model quality, it also has significant potential to enhance model fairness, safety, and reliability. For instance, metrics designed to quantify various forms of bias can aid in identifying and addressing model unfairness. While such metrics exist for classification or regression models (e.g., demographic parity or equalized odds), their extension to generative models is non-trivial. Research is thus needed to develop and refine metrics that can effectively quantify biases in the complex outputs of generative models.[8] This includes creating frameworks that account for the nuanced and context-dependent nature of generated content, ensuring these models are not only high-quality but also fair and aligned with ethical standards (Ray, 2023). Unfortunately, to be effective, these metrics must also be adaptable to closed-source generative models since parameters and logits of most commercial models are not publicly available (Zhao et al., 2023a; Sun et al., 2023a; Laszkiewicz et al., 2024).

Despite the availability of more advanced evaluation metrics, some domains continue to rely heavily on outdated automatic evaluation methods. For instance, BLEU (Papineni et al., 2002) and ROUGE (Lin, 2004)—metrics based on n-gram matching that are known to empirically correlate poorly with human judgments (Reiter, 2018; Deutsch et al., 2022)—remain extremely prominent in the evaluation of machine translation and abstractive summarization systems. The continued reliance on these inaccurate metrics may ultimately impede the advancement of generative AI, as they provide weak signals for model improvement and fail to guide the development of systems that truly align with human expectations and real-world applications. A shift towards new evaluation metrics requires a critical mass of adoption within the community. Therefore, we must encourage practitioners to move beyond the convenience of outdated metrics and embrace this new generation of improved metrics.

## 4 Ethical Deployment and Societal Impact

With the current excitement around the scope and application of large-scale generative models, we are also witnessing a growing apprehension, fueled by media reports, of adverse outcomes surrounding the rapid advancement of generative AI. These concerns add to the conceptual and practical considerations discussed so far and encompass a range of issues, including the spread of misinformation, the absence of regulatory frameworks (Meskó & Topol, 2023), unintended harm (Greenfield & Bhavnani, 2023), and debates over open-source versus closed-source technologies (Chen et al., 2023), among others. Here we identify key challenges concerning the responsible deployment of large-scale DGMs. More specifically, we discuss several aspects, including the dissemination of misinformation (4.1), violation of privacy and copyright (4.2), presence of biases (4.3), lack of interpretability (4.4), and constraint satisfaction (Section 4.5).

---

[8]Many aspects of fairness cannot be captured by quantitative metrics. Further, definitions of fairness can differ amongst different people and groups, and these definitions may evolve over time. However, they can still provide insights into whether models achieve a certain level of fairness in specific aspects.

| Sub-challenge | Typical failure mode | Promising mitigation avenues | Reference materials | Research-question candidates |
|---|---|---|---|---|
| Misinformation & synthetic media detection (§4.1) | Deepfakes bypass detectors | Model-rooted watermarks (e.g., Tree-Ring Watermarks) | **Datasets:** Deepfake Detection Challenge ; WaterBench **Surveys:** Rana et al. (2022) | *Q1*: Can diffusion-time watermarking survive multimodal adversarial attacks? *Q2*: Can uncertainty-aware abstention policies mitigate hallucinated facts in DGM outputs? |
| Privacy violations, copyright infringement (§4.2) & PII leakage | Model regenerates training data verbatim or near-verbatim | Differential Privacy learning techniques (e.g., DP-SGD); Machine Unlearning methods | **Datasets:** CPDM; PrivLM-Bench **Surveys:** Yao et al. (2024) Kibriya et al. (2024) | *Q1*: What privacy-preserving fine-tuning strategies remain feasible for $\geq$ 100B-parameter models under practical compute budgets? |
| Fairness across languages (§4.3) | Worse compression $\rightarrow$ more tokens needed $\rightarrow$ higher cost for low-resource languages | Dynamic tokenization schemes; Vocabulary transfer methods | **Surveys:** Xu et al. (2025) Qin et al. (2025) | *Q1*: Can subword-free sequence modeling architectures (e.g., SSMs) eliminate tokenization-induced performance disparities across languages? |
| Bias & discrimination in generated content (§4.3) | Models reflect and propagate bias and discrimination present in training data | Counterfactual evaluation benchmarks; Fairness metric (e.g., demographic parity) incorporation into training objectives | **Datasets:** HolisticBias; OpenBias **Surveys:** Gallegos et al. (2024) | *Q1*: How do watermarking methods interact with the appearance of demographic biases in model outputs? |
| Interpretability & transparency (§4.4) | DGM parameters are uninterpretable, making them black boxes for users and developers | Automated circuit discovery; Causal tracing | **Datasets:** GPT-2 neuron-explanation dataset **Surveys:** Marcinkevics & Vogt (2020) | *Q1*: Which confidence metrics best predict whether a mechanistic explanation truly modulates model behavior? *Q2*: Can mechanistic insights be transferred across model sizes? |
| Constraint satisfaction (§4.5) | Outputs violate hard rules (e.g. code won't compile) | Context Free Grammar-guided decoding algorithms; Constrained RLHF; Constraint-embedded model architectures | **Benchmarks:** HumanEval; BigCodeBench **Surveys:** Zhang et al. (2023a) | *Q1*: How can hard-constraint decoding be generalised from code to text and images? *Q2*: Do constraint-aware training methods (e.g., constrained RLHF) achieve better safety-performance trade-offs than inference-time constraint enforcement (i.e., constrained decoding)? |

Table 3: Research challenges and mitigation strategies - Ethical Deployment and Societal Impact

## 4.1 Misinformation and Uncertainty

As the quality of generated data synthesized using large-scale generative models increases, it can become more and more difficult to distinguish between real and generated content, especially for uninformed consumers (Frank et al., 2024). This indistinguishability facilitates the spread of misinformation (e.g., by deepfakes (Helmus, 2022)). To ensure the trustworthiness of information, we need algorithmic solutions that are on par with the advances in generative models and allow us to robustly detect and mark synthetic data. Numerous models for differentiating machine-generated from real content have been proposed over the last years (Rana et al., 2022), but the increasing quality of generative model outputs has decreased their accuracy. Watermarking is another approach in which there has recently been increased interest. The goal of these methods is to manipulate a generated sample (e.g., an image or piece of text) such that a signature can be detected in downstream tasks, albeit with minimal effects on sample quality. There have been several approaches to watermark synthetic data generated from LLMs (Kirchenbauer et al., 2023; Dathathri et al.,

2024; Zhao et al., 2024) and image generation models (Zhao et al., 2023c; Wen et al., 2023; Jiang et al., 2024b). However, current watermarking methods are far from robust (Saberi et al., 2023). Evasion is often possible by small manipulations (like paraphrasing or pixel perturbations) (Jiang et al., 2023) and the injecting of watermarks into content can be inefficient (Liu et al., 2024a). Recent work has demonstrated that model-integrated watermarks—i.e., signatures embedded in a model's sampling process rather than post-hoc in its outputs—are a promising route forward, as they can survive a range of real-world corruptions (e.g., paraphrasing attacks in text (Krishna et al., 2023) and latent-trajectory perturbations in images (Wen et al., 2023; Fernandez et al., 2023)). Nonetheless, the information-theoretic limits on detectability, false-positive rate, and adversarial removability for watermarking remain poorly understood, and initial negative results suggest unavoidable trade-offs between robustness, the quality of the altered sample, and watermark payload (Kirchenbauer et al., 2024; Yoo et al., 2024).

Notably, misinformation can emerge even without malicious intent. Tools like ChatGPT are increasingly expected to serve as universal question-answering engines, even though their core objective—to estimate the likelihood of the next token in a sequence—is traditionally designed to assess the linguistic plausibility of strings (Kalai & Vempala, 2023), rather than their factual accuracy. This distinction between the two objectives is evinced by the discrepancies observed between the probability a model explicitly assigns to a statement when prompted vs. the underlying log-probability it assigns (Hu & Levy, 2023), models' tendencies to hallucinate (Huang et al., 2024a), and their difficulty in achieving probabilistic consistency (Elazar et al., 2021), e.g., ensuring logical predictions between a statement and its negation.

Some works have turned to model uncertainty estimates (e.g., those discussed in 3.2) as indicators of model reliability, developing methods to enhance the trustworthiness of AI systems based on these estimates (Edupuganti et al., 2021; Yang et al., 2023e). For example, Ren et al. (2023) propose a selective generation approach, where models abstain from providing a response in the face of high uncertainty; Kuhn et al. (2023) use a notion of a model's semantic uncertainty to predict the correctness of its answer in question-answering. The development of methods that explicitly account for uncertainty represents an interpretable approach toward ensuring model reliability, offering a strategy that also has grounding in a well-studied concept in machine learning (Malinin & Gales, 2018; Abdar et al., 2021; Gawlikowski et al., 2023).

Encouragingly, some research suggests that larger LMs actually are well-calibrated in terms of their world knowledge, i.e., their predicted likelihoods reflect the probability that a statement is true (Srivastava et al., 2022; Zhu et al., 2023; Yu et al., 2024). Further, recent studies show that language models often do possess the ability to assess the truthfulness of their own statements (Lin et al., 2022; Kadavath et al., 2022; Xiong et al., 2024). However, fine-tuning or RLHF, which are frequently applied to these models, have been shown to hurt calibration; rather, they have been widely observed to exhibit overconfidence—the tendency of a model to assign excessively high probabilities to its predictions regardless of their correctness (Kadavath et al., 2022; Tian et al., 2023; OpenAI, 2023; Xiong et al., 2024). There has been some work on mitigating miscalibration issues for fine-tuning (e.g., Wang et al., 2023b) and RLHF (e.g., Tian et al., 2023; Zhang et al., 2024), and there is an increasing focus on systems where evidence can be brought in from external knowledge sources (Blattmann et al., 2022; Pan et al., 2023; Gao et al., 2023), grounding model responses to reliable knowledge sources. Such research—along with benchmarks to assess model factuality (e.g., KoLA; Yu et al., 2024)—is a critical step towards ensuring the trustworthiness and reliability of generative models.

## 4.2 Security, Privacy and Copyright Infringement

While modern generative models like LLMs are deployed practically for many applications, this also exposes them to potential malicious attacks, which can have significant costs for downstream applications or users. One class of malicious attacks on LLMs is the so-called "backdoor attacks" where the main idea is to train the model using *poisoned data* and then trigger a specific output response from the model corresponding to specific prompts. For instance, Yang et al. (2023a) discusses backdoor attacks on LLMs in the context of communication networks (see Zhou et al., 2025, for a more in-depth exploration of backdoor attacks). Another class of attacks known as "jail-breaking" involves designing adversarial prompts to generate malicious outputs from the model while bypassing guardrails employed to comply with usage policies. There has been a good deal of research in the context of LLMs exploring techniques for jail-breaking and for guarding against jail-breaking (Shen et al., 2024; Yi et al., 2024; Jin et al., 2024).

Beyond malicious attacks, another critical risk associated with generative AI models is their tendency to memorize and unintentionally reproduce training data, raising serious concerns about privacy and data leakage. A variety of works have shown that publicly available LLMs and large-scale text-to-image models can implicitly "memorize" training data (van den Burg & Williams, 2021), to the point that samples from the dataset can be (almost exactly) reconstructed (Carlini et al., 2023a;b; Somepalli et al., 2023; Nasr et al., 2023). This behavior potentially infringes on data privacy, underscoring the importance of detecting whether private information has been leaked into an LLM's training data (Kim et al., 2023) and exploring whether generative models can be trained while safeguarding sensitive information. Differential privacy (DP) constraints, which can be enforced during generative model training, offer an attractive theoretical framework to ensure privacy Dwork & Roth (2014); Li et al. (2021); Dockhorn et al. (2023). However, DP-based approaches have several shortcomings. They suffer from a trade-off between privacy and utility (Cummings et al., 2024). There are different DP formulations, each based on different assumptions about trust, data access, and the point at which noise is introduced. The meanings of the canonical DP parameters are thus not consistent, making comparison of models produced using different approaches difficult (Li et al., 2024d). Recent work has instead focused on generative DP synthetic data, where foundation models are only used as black boxes (Lin et al., 2024b). Building privacy constraints into the training of large-scale generative models can be a promising direction for further research.

Another byproduct of memorization in generative models is that it can lead to unauthorized distribution or replication of training data, resulting in copyright infringement liabilities.[9] Current efforts towards preventing such behavior have approached the problem from different angles. Some focus on filtering training data: techniques such as contractual licensing filters and hash-based de-duplication allow developers to identify and exclude protected material before training (Carr & Jeffrey, 2022; Duarte et al., 2024). Plagiarism or style-clone detectors allow intervention downstream, flagging generated samples that are substantially similar to copyrighted works (Li et al., 2024c; Kim et al., 2021). Other works have proposed training methods, which can be broadly categorized under two complementary strategies: (i) *imitation-resistant training objectives* that discourage verbatim memorization (Liang et al., 2023; Zhao et al., 2023d), and (ii) *machine unlearning* methods that aim to eliminate the influence of certain datapoints (e.g., copyrighted or private material) from a model's predictions after training (Li et al., 2024b; Liu et al., 2024b; 2025). There is still much open research in this domain, including provenance tracking that can scale to work with modern, massive datasets and copyright infringement risk metrics that encompass legal concerns.

Beyond privacy and copyright concerns, indiscriminate memorization has other undesirable effects. Many of the current target applications for generative models—such as creative writing or graphics generation—demand novelty and user-specific adaptation; when a model merely regurgitates training data, it does not provide the desired diversity, originality and personalization. This behavior potentially worsens user experience and limits the practical value of generative models. Further, excessive memorization can lead to biases in generated content, where certain perspectives or styles dominate because they were overrepresented in the training data; we discuss this last issue in more detail in the next section. Ultimately, deploying DGMs safely and usefully will require turning the request of "don't copy the training set" into an explicit design and auditing objective, which necessitates continued research into how to measure, control, and (when necessary) unlearn memorized examples (Chen et al., 2024a; Lesci et al., 2024).

### 4.3 Fairness

Large-scale generative models are often trained on massive datasets containing billions of samples scraped from the internet. While preprocessing such large datasets often involves tagging or removing toxic content, a variety of other societal biases are often harder to detect. Consequently, the trained models can reflect biases and produce outputs that may be deemed toxic or harmful (Pagano et al., 2023; Gallegos et al., 2024; Zhou et al., 2024). For instance, Weidinger et al. (2021) outline a series of harms that can result from using LLMs that produce discriminatory or exclusionary language, e.g., the amplification of stereotypes or exclusionary norms. Multimodal models may exhibit biases about gender, ethnicity, and religion, among others (Janghorbani & De Melo, 2023) that have similar negative effects on society.

---

[9]https://www.nytimes.com/2023/12/27/business/media/new-york-times-open-ai-microsoft-lawsuit.html

Not all unfair behaviors exhibited by generative models are as overtly harmful as generating toxic content. Models can show more subtle biases towards certain subpopulations, such as allocation bias—when AI systems extend or withhold opportunities, resources, or information—or quality-of-service bias—when AI systems work better for people in some subpopulations than others; these behaviors should not be downplayed as they can perpetuate systemic inequalities. A main culprit of such behaviors stems from the fact that the data used in the training of most generative models is disproportionately from certain countries and languages. Such models therefore might not work as well for languages or images outside of these mainstream groups. For example, multilingual language models typically perform substantially better on English tasks compared to tasks in other languages (Lai et al., 2023; Huang et al., 2023a). Further, because the tokenizers for such models have also been trained disproportionately using English data, the compression rate for English texts is much higher than for texts in low-resource languages (Petrov et al., 2023; Ahia et al., 2023). Consequently, services that charge based on token counts impose higher costs for a query made in a low-resource language compared to a query with the same underlying meaning made in English. This exacerbates accessibility challenges for speakers of underrepresented languages.

Numerous approaches have been proposed to mitigate the biases of generative models in a post-hoc manner (Bai et al., 2022; Glaese et al., 2022; Ferrara, 2024; Olmos et al., 2024). However, the achieved changes are often merely superficial, leaving the possibility of remnant biases. For example, Gonen & Goldberg (2019) demonstrate that word embeddings still cluster based on gender stereotypes even after bias mitigation techniques, effectively "hiding" rather than eliminating the issue. In the vision domain, post-hoc bias mitigation strategies have been observed to work poorly in the face of test-time distribution shift (Kong et al., 2023). Moreover, most evaluations assess only a *single* fairness axis—for instance, gender in English or skin-tone in photos—and residual biases along other dimensions can remain undetected. Some key research areas that we identify as needing further attention include: (i) joint evaluation across multiple, potentially interacting fairness criteria (e.g., gender × dialect); (ii) stress-testing mitigation strategies under data-distribution shifts; and (iii) training-time interventions that prevent harmful biases from emerging in the first place. Some areas that we identify as needing further research are combined evaluation with respect to multiple forms of fairness criteria, robust assessments across multiple domains and training methods that can more robustly mitigate the learning of harmful biases. Promising steps in these directions include multilingual, multi-attribute benchmark suites such as XCOPA-Bias (Goldfarb-Tarrant et al., 2023) and gradient-based de-biasing schedules that adjust sampling weights during pre-training (Kim et al., 2024).

Ultimately, assessing and ensuring fairness in technology applications is a complex challenge. Aside from the aforementioned issue of differing (and potentially dynamic) qualitative definitions of fairness (3.2), different notions of fairness often cannot be fully satisfied simultaneously (Ferrara, 2024). Therefore, it is essential for the builders of generative AI tools to carefully evaluate the various dimensions of fairness and make deliberate trade-offs appropriate for the specific use case.

## 4.4 Interpretability and Transparency

In high-stakes applications such as healthcare and legal domains, it is critical to understand the logic and influencing factors behind generative models' outputs. In other words, we need to be able to *interpret* how a generative model produces its results, with its decision-making process being *transparent*, i.e., accessible and understandable This is particularly true in safety-critical domains—such as healthcare (Chen et al., 2021a) or finance applications—but is also important across general AI use cases, where interpretability is essential for diagnosing errors and fostering user trust. These needs are not new and have been present since the start of publicly available AI-based products and tools (Confalonieri et al., 2021). There has thus been a sizable amount of research in neural network interpretability methods. However, these methods are not always feasible for use with large-scale DGMs. For example, interpretability methods, such as SHAP, LIME, or Integrated Gradients (Lundberg & Lee, 2017; Ribeiro et al., 2016; Sundararajan et al., 2017), struggle to scale effectively with the complexity and size of large models; many interpretability methods work by attempting to understand concepts encoded in models' latent representations (Crabbe et al., 2021; Esser et al., 2020), but this becomes more difficult in the high-dimensional latent spaces used by DGMs. Further, while one might hope that explanations derived from interpreting smaller models could be used for understanding their larger counterparts, scaling up generative models gives rise to unpredictable effects,

e.g., models demonstrating unexpectedly advanced capabilities (Wei et al., 2022; Schaeffer et al., 2024); conclusions drawn when using small models therefore may not apply to today's larger models.

The fundamental challenge is to develop explanation methods for DGMs that are both well-understood by humans and faithful to the underlying model behaviors (Schut et al., 2023; Gurnee & Tegmark, 2024). *Mechanistic interpretability* is a field that attempts to achieve this goal by reverse-engineering neural network decisions, translating them to human-interpretable decision-making processes (Bereska & Gavves, 2024). This is specifically done by analyzing models at the level of their internal computations, representations, and structural components, e.g., identifying minimal subnetworks (referred to as circuits) that implement a specific computation. A large appeal of mechanistic interpretability is that it provides causal explanations for models' outputs, i.e., a decision or prediction can be attributed in a causal manner to some component of the input. For example, causal tracing, a prominent tool in this line of work, perturbs internal activations in a manner that allows us to determine whether they *cause* a change in the output. This allows researchers to move beyond correlation-based explanations and instead understand the actual computational mechanisms driving observed model behavior, which in turn enables more precise debugging, bias detection, and control over generative outputs. Mechanistic interpretability research has uncovered a number of interesting and useful properties of DGMs. For example, specific neurons or layers in GANs and VAEs encode "disentangled" (i.e., distinct and interpretable) features, such as shape, texture, or pose in images (Shen et al., 2020; Mita et al., 2021). Other works have found that certain attention heads in transformers correspond to meaningful linguistic patterns, e.g., some might focus on syntactic structure while others might capture semantic information (Vig & Belinkov, 2019; Elhage et al., 2021). Such properties not only help practitioners better understand DGMs, they also enable them to control generation to some extent (Härkönen et al., 2020; Fetty et al., 2020).

However, the reliability and comprehensiveness of mechanistic interpretability methods remain a subject of debate (Golechha & Dao, 2024a; Sharkey et al., 2025). A central criticism is that, while these techniques aim to identify causal relationships between model components and outputs, they may only provide partial or even misleading insights into the actual computational processes at work. For instance, attention patterns— which have been used by various methods to attribute model predictions to certain tokens (Xu et al., 2015; Choi et al., 2016, *inter alia*)—do not always faithfully reflect how or why certain tokens influence the final prediction (Jain & Wallace, 2019; Liu et al., 2022b). In some cases, they may merely highlight correlations rather than reveal deeper causal structures. Similar concerns have been raised about other methods for explaining model behaviors from mechanistic interpretability. Further, the new wave of large-scale generative models makes the application of some of these methods more difficult: the larger a model becomes, the (arguably) more difficult it becomes to fully reverse-engineer a prediction, both from a theoretical and computational standpoint. Every nuance of these models' decision-making may not be deducible from a subset of neurons, layers, or attention heads, and interpretations derived from one subset of model components may overlook equally critical interactions elsewhere in the large network.

Representation engineering (Zou et al., 2023a) and the use of sparse autoencoders (Bricken et al., 2023) are lines of research in mechanistic interpretability that potentially address the former issues, offering interpretable explanations even for large-scale models. Efficient methods for circuit identification have started to address the latter issue (Hsu et al., 2025). However, we are still in need of validation methods that can confirm whether the identified "mechanisms" truly govern a model's outputs, or whether they merely reflect convenient, yet incomplete, narratives about its internal workings.

Going forward, researchers should continually assess user needs for explainability to ensure that the appropriate objectives are guiding the development of interpretability methods (Liao et al., 2020; Wang & Yin, 2021; Poursabzi-Sangdeh et al., 2021). Attention must also be paid to the relevance and effectiveness of the metrics and evaluation frameworks used to assess these methods (Ross et al., 2021; Jethani et al., 2021). Another important research direction is enhancing the robustness of explainable methods, such as counterfactual explanations (Wachter et al., 2017; Slack et al., 2021). Further research is also needed for the new wave of multimodal models, as existing explainability methods may not be equipped to offer explanations in the face of cross-modal interactions.

### 4.5  Constraint Satisfaction

Generative models such as ChatGPT are used by millions of people and deployed across diverse use cases. Many applications require generative models to satisfy domain-specific constraints.[10] In some cases, these merely stem from a desire to have a more controlled form of generation, such as when a generated image is conditioned on a given depth map (Zhang et al., 2023b). In other cases, ethical and safety considerations are key concerns. For instance, in fields like engineering design, generative model outputs must meet engineering standards and adhere to laws of nature (i.e., physics). More generally, there are widespread calls for generative models to avoid toxicity, mitigate bias and prevent other outputs that may lead to harmful effects (Weidinger et al., 2021), e.g., by refraining from responding in ways that could pose a risk to the mental health of human interlocutors and by refusing to carry out tasks related to illegal activities. While reinforcement learning from human feedback (RLHF; Ouyang et al., 2022)—and particularly *constrained* RLHF (Moskovitz et al., 2024)—has offered an initial step towards these goals, enabling companies and users to provide models with soft constraints within their queries, such constraints can be circumvented (Shen et al., 2023). Ultimately, methods that allow us to place hard constraints on model outputs are necessary.

In language generation, decoding methods that allow for arbitrary constraints (both hard and soft) have been a focus area of the research community (Kumar et al., 2021; 2022). There are several prominent challenges in the development of such methods, including: efficiently enforcing constraints without significantly increasing computational costs, maintaining fluency and coherence while adhering to constraints, and handling multiple constraints, which may result in conflicting requirements. The discrete nature of text presents a particular difficulty, as small changes to token sequences can drastically alter meaning, making it difficult to optimize for constraints from a computational perspective. Recent grammar-constrained decoding methods (Beurer-Kellner et al., 2024; Ugare et al., 2024) address these issues by guaranteeing that every generated sequence conforms to a user-supplied context-free grammar, achieving hard constraints with only a modest runtime overhead. Such methods have started to gain traction in other domains, e.g., in healthcare (Golechha & Dao, 2024b). Meanwhile, research on controllable image generation has likewise gained momentum (Deng et al., 2020; Huang et al., 2024b), with various approaches aiming to regulate attributes such as style, composition, or specific content elements. Methods range from applying spatial constraints (e.g., bounding boxes, masks, or layout specifications (Zheng et al., 2023a)) to enforcing semantic conditions (e.g., ensuring that certain objects or visual features are present (Pavllo et al., 2020)). Technical challenges arise in this domain as well, such as the need for more complex conditioning mechanisms and heavier computational demands—especially when constraints must be integrated at each step of the generative process in e.g., diffusion models. Code generation stands out as a domain where constraint enforcement has long been a primary focus (Poesia et al., 2022; Dong et al., 2023). Here, the constraints—such as following a language's syntax and producing compilable code—are arguably more well-defined and straightforward to verify. Concepts and methods from this field could conceivably help research in enforcement of hard constraints in other generative AI fields, e.g., in the application of generative AI to the physical domain, where laws of nature must be satisfied. Overall, methods to ensure effective constraint adherence can substantially improve control over generative models, which is crucial for ensuring their safe and reliable deployment (Regenwetter et al., 2024). Despite progress across various generative AI fields, the development of scalable and generalizable techniques capable of handling the diverse and often conflicting demands of real-world constraints remains an open challenge. We encourage further research in this area to bridge this gap and advance the controllability of generative models across different domains.

## 5  Conclusion

Generative AI has achieved remarkable progress in recent years, pushing the boundaries of what computational systems can model and generate. Yet, this rapid evolution also comes with a wide array of technical, ethical, and societal challenges that demand careful attention. Thus, the goal of achieving a *perfect* generative model remains far from reality. As we have outlined throughout this paper, current generative AI models

---

[10]Here we focus on constraints specified at inference time. We discuss constraints that must be integrated into the model during training in 2.3.

struggle with robustness under distribution shift, remain resource-intensive, and often exhibit behaviors that are difficult to interpret or control, particularly in high-stakes and specialized domains.

Crucially, the path forward is not merely a matter of scaling existing paradigms. Despite gains in performance from ever-larger models and training corpora, core limitations persist. Addressing them requires (re)consideration of foundational assumptions. We have emphasized the importance of integrating causal reasoning into model architectures to overcome spurious correlations and improve generalization, especially under distributional shifts. Likewise, embedding domain knowledge and inductive biases can enable more data-efficient learning in fields like healthcare, chemistry, and the physical sciences, where data is often limited but expert insight is abundant.

Efficiency and accessibility are also central concerns for facilitating the widespread use of DGMs. Today's DGMs are not only expensive to train and deploy but also have created a growing barrier to entry for researchers and practitioners without access to large-scale compute. Optimizing inference efficiency through quantization, architectural innovations, and fast sampling methods is essential for democratizing generative AI and minimizing its environmental footprint. At the same time, robust evaluation remains an open challenge: current metrics often fail to align with human judgments or to meaningfully assess qualities such as diversity, calibration, or ethical compliance. The development of more principled and context-dependent evaluation frameworks—potentially learned from human preferences—will be vital to guiding model development and ensuring model trustworthiness.

Beyond technical concerns, we must also address the broader societal and ethical impacts that generative AI can have. Interpretability, fairness, and safety must become a priority in design considerations, not an afterthought. These systems have already demonstrated the ability to generate misinformation, propagate social biases, infringe on privacy, and produce unsafe or unreliable outputs. While recently-developed mitigation strategies, from watermarking and uncertainty-aware abstention to differential privacy and constraint-aware decoding, appear to offer promising solutions, the methods alone are insufficient without regulatory frameworks for ensuring their usage.

We believe that addressing this range of issues will require deeper cross-disciplinary collaboration, new benchmarks that reflect real-world complexity, and an intentional shift from optimizing for performance to optimizing for positive societal impact. By confronting the limitations discussed here, we can transform DGMs from data replicators (Bender et al., 2021) to tools with transformative capabilities across various domains. We hope that the information and opinions presented here will point to directions that ultimately contribute to these goals.

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
