# OpenReview forum: "On the Challenges and Opportunities in Generative AI"
_TMLR — Accepted by TMLR_

### Review · Reviewer_pcoR · 2025-04-08

**Summary Of Contributions:**

The paper discusses a range of challenges in deep generative models, including the following
- generalization and adversarial robustness,
- implicit and explicit assumptions, priors, and inductive biases
- integration with causal models
- cross-modality, integration of domain knowledge
- efficiency and quantization
- meaningful metrics, aligned with human evaluation
- uncertainty quantification and hallucination
- privacy, fairness, explainability, and constraint satisfaction

Across all these topics, the paper summarizes key challenges and provides extensive references to the literature.

**Audience:**

Yes

**Claims And Evidence:**

Yes

**Requested Changes:**

I would suggest that the authors consider the points raised under Weaknesses and make any necessary revisions. In its current form, I believe the paper satisfies the criteria for publication, and would be useful to some readers looking to get an overview of the field and a curated set of references.


A few typos etc.

"...(Grattefiori et al., 2024). have enabled..." dot after parenthesis

"... resilient to minor variations..." Is this a precise definition of adversarial robustness?

"...with the provided description. ." two dots

"For example, (Ren et al., 2023)..." -> "For example, Ren et al. (2023)..."

"(Kuhn et al. 2023) use a..." -> "Kuhn et al. (2023) use a..."

**Strengths And Weaknesses:**

Strengths:
- The set of problems addressed in the paper are important and very timely.
- The paper gives a broad (although somewhat superficial) overview of several important topics within deep generative models.
- The paper includes many (around 375) references, and may serve as a good resource to navigate the literature.
- The writing is clear and accessible.


Weaknesses:
- The paper does not present any new findings, novel insights or methodologies, but only summarizes known results.
- The content selection in the review does not appear to be selected using a systematic approach.
- The contents could be structured better, perhaps by summarizing key points, providing overviews in the form of tables or figures, or by providing clear, actionable recommendations.
- The centrally defined "perfect generative model" in the introduction remains elusive, being either under-specified or, at face value, seemingly impossible.
- The paper has a very broad focus, exemplified primarily through language and image models. It is not clear what the intended audience is.
- Overall, the discussion remains quite abstract, which can occasionally make it challenging to understand how the discussed challenges could be addressed in a practical context. As a result, the reader gains more of an overview than a deep understanding.

Many of the conclusions throughout the paper remain very general, which makes it difficult for the reader to contextualize. One example is:

"We argue...explicity encode human priors into the training process." How that could be done in a practical setting is not clear. Perhaps it would be more clear with some well chosen examples.


There are some statements throughout that seem overly general and perhaps not sufficiently substantiated. I will just mention a few examples:

"...inductive biases, largely driven by convenience or convention. (...) We argue that convenience should not be the driving factor..." This seems to me to be a too strong statement. Many inductive biases are well thought through and well understood, such as translational equivariance in CNN's etc. What exactly is meant by convenience here? I would argue that practicality (i.e. what works) is a stronger driving factor than convenience or convention.

"The overarching goal is (...) applications (...) across different types of data (...) embodied agents must integrate perception..." I do not follow the argument that applying DGM across domains necessitates embodied agents and addressing control tasks.

---

> ### Author Response · Authors · 2025-05-30
> **Author Response**
>
> Thank you for your thoughtful and constructive feedback. We appreciate the great attention to detail you gave to the manuscript and for highlighting both its strengths and the areas where it can be improved. Below we address each of your comments in turn and outline the steps we have taken in the revision.
>
> **Lack of novel findings or methodologies**:
> Our goal was to deliver a summarizing survey that maps the fast-moving terrain rather than to introduce new algorithms or techniques. Rather than novel findings, we believe that our paper offers insights, which can be just as valuable. We have added three summary tables (one per core part) with “Promising mitigation avenues” and “Research-question candidates” columns in the hopes of more concretely providing insights.
>
> **Need for clearer structure, summaries, and actionable recommendations**:
> We agree that long prose can overwhelm. We hope that our summary tables add some of the needed structure, outline the main points that we discuss and highlight actionable recommendations, which hopefully provides what the reviewer finds missing.
>
> **Elusive definition of the “perfect generative model”**:
> We agree that the original metaphor fails to give operational meaning. We have added a more concrete definition for the context of the paper, clarifying that it is not comprehensive but is what the authors find to be an important subset of criteria.
>
> **Unclear intended audience & breadth vs. depth trade-off**:
> Our aim is to serve researchers and advanced practitioners who need a broader understanding of the field. We focus on image and language models because these are the most prevalent, and the vast majority of research across different domains builds off of these models in some respect. We have revised the introduction to better pinpoint the targeted audience.
>
> **Discussion too abstract; need practical examples**:
> We agree that concrete cases ground the discussion. We will work to incorporate these more throughout the manuscript.
>
>
> **Over-general statements (inductive biases, embodied agents)**:
> We will carefully go through the paper and try to identify such statements, adding clarifications and citations. With respect to the specific instances that you mentioned:
> **Inductive biases.** This is a fair point. We have toned down the claim and explicitly list “well-understood” biases (e.g., translational equivariance) vs. convenience-driven defaults.
> **Embodied agents.** We have clarified this argument.
>
> **Adversarial-robustness wording**:
> This is not a formal definition and we have clarified that “resilient to minor variations” is just a criterion loosely related to the more formal notion of adversarial robustness
>
> **Further minor edits**:
> We will perform a multiple proof-reads and correct misplaced parentheses and citation styles.

---

> > ### Comment · Reviewer_pcoR · 2025-06-03
> >
> > Thank you. As mentioned, I believe the paper meets the criteria for publication. It has improved with the revision, and although some weaknesses remain unaddressed, my positive recommendation remains unchanged.

---

### Review · Reviewer_pxE1 · 2025-04-24

**Summary Of Contributions:**

The paper covers a wide range of currently open research problems in the field of generative AI. These include the areas of generalization, robustness, efficiency, evaluation, interpretability, fairness, and societal impacts, among others. In addition to reporting on these challenges and recent research attempting to solve them, the authors highlight certain directions which they believe would particularly benefit the community if solved. The result is a comprehensive review of the current challenges in generative AI, which should aid researchers looking to help solve these problems.

**Audience:**

Yes

**Broader Impact Concerns:**

No concerns.

**Claims And Evidence:**

Yes

**Requested Changes:**

# Security concerns

Security concerns such as jailbreaking and backdoor attacks could significantly affect adoption in critical systems with a generative AI component, particularly when using LLMs. A discussion of the research in this area would strengthen the work.

For example, the following are useful starting points for references for backdoor attacks:
- "A Comprehensive Overview of Backdoor Attacks in Large Language Models within Communication Networks"
- "A Survey on Backdoor Threats in Large Language Models (LLMs): Attacks, Defenses, and Evaluations"

And for jailbreaking:
- "Jailbreak Attacks and Defenses Against Large Language Models: A Survey"
- ""Do Anything Now": Characterizing and Evaluating In-The-Wild Jailbreak Prompts on Large Language Models"

# Environmental impact

It would be useful to have an indication of the environmental impact of large models, beyond high energy costs. What are the estimated carbon emissions of these models? What percentage of electricity usage does generative AI currently make up and how much is it projected to make up in the future? Answers to these questions would improve the submission, since environmental impact is also a social cost and a major challenge going forward.

# Minor mistake

On page 10, the reference to Xu et al. 2020 is incorrectly formatted. The "e.g." should come before the reference or be dropped.

**Strengths And Weaknesses:**

# Strengths
- Coverage is fairly comprehensive
- The manuscript is well structured and clearly written. I never had trouble understanding what the authors meant
- A lot of attention is given to evaluation metrics, which I find justified, given their importance for the progress of the field
- Discussion is detailed and well contextualized within the current literature

# Weaknesses
- The discussion of robustness does not cover security concerns, e.g. jailbreaking
- There is no real discussion of environmental impact of models

---

> ### Author Response · Authors · 2025-05-30
> **Author Response**
>
> We thank the reviewer for their constructive feedback. Please find our response below based on the changes requested by the reviewer
>
> Requested Changes:
>
> Security Concerns: We thank the reviewer for highlighting this line of work and suggesting references. We have updated Section 4.2 in our manuscript to highlight some of these concerns in the context of LLMs.
>
> Environmental Impact: We agree that there might be environmental concerns associated with the large-scale training of generative models. However, we believe there are contrasting narratives on the same (for instance, see https://www.nature.com/articles/s41598-024-76682-6), and more research is needed to probe into the sustainability of training large-scale language models. We have therefore actively chosen not to discuss this aspect in detail in the paper, but have added a pointer for interested readers.
>
> Minor Mistake: Thanks for pointing this out. We have fixed this in the revised version of our manuscript.

---

> > ### Comment · Reviewer_pxE1 · 2025-06-06
> >
> > Thank you for making the requested changes. With these changes I can recommend acceptance.

---

### Review · Reviewer_HKXq · 2025-04-28

**Summary Of Contributions:**

The paper proposes a comprehensive overview of the current shortcomings of recent generative AI methodologies, highlighting the related challenges and respective areas where further research is needed. In this regard the authors explore the topics of generalization to specific domains, robustness to malicious or out-of-distribution inputs, prior knowledge incorporation, causal modeling, handling of heterogenous data, other than text and images. Moreover also topics related to computational efficiency and model evaluation are explored, as well as ethical and societal aspects.

**Audience:**

Yes

**Claims And Evidence:**

Yes

**Requested Changes:**

- I would suggest that the authors state more clearly what differentiates their work from other surveys that discuss similar topics.
- I think it would meaningfully improve the paper if the authors could concisely provide references to topic-specific surveys (or the most meaningful works, in their absence); for instance by means of a table. As the scope of the discussion is very broad, this could provide a structured point of aggregation for the reader that wants to read further on specific topics, even though these references are already provided organically along the document.

**Strengths And Weaknesses:**

**Strengths**
The paper is very well written and easy to read. It includes a diverse number of aspects related to existing challenges in generative models, provides relevant references and, in some cases, highlights possible paths or hints to address the mentioned challenges.

**Weaknesses**
As the paper explores many diverse topics in a relatively small number of pages it keeps the discussion on an high level, without delving into much details for most of the challenges it mentions. While this makes it a useful and comprehensive read for someone approaching the field, I worry much of the provided information might be already known to researchers in the area (at least at this depth of a discussion level).

Moreover there is no direct discussion regarding the surveys that already exist about the topic or sub-topics. On one hand this makes it hard to objectively evaluate the usefulness of this paper compared to the existing literature. On the other it is a lost opportunity to reference, in a structured and concise way, existing studies that, while maybe more limited in scope, delve deeper into the details of specific sub-areas related to generative models.

---

> ### Author Response · Authors · 2025-05-30
> **Author Response**
>
> We thank the reviewer for the constructive feedback. We have revised the introduction to further highlight the goal of our paper and the targeted audience. Specifically, we emphasize how this paper serves as an integrated entry point for researchers and practitioners, specifically newcomers to the field. Our goal is not to provide a deep dive into specific subtopics, but rather to offer a high-level overview that synthesizes key insights across the full spectrum of generative AI research to guide future research toward what we perceive are the most critical and promising areas.
>
> To further strengthen the paper and support the reader’s ability to navigate the literature, we have also added a set of structured tables, which consolidate key references to topic-specific surveys and foundational works. These tables serve both as a navigational aid and as a curated entry point for readers who wish to explore particular challenges in greater depth.
>
> We believe these additions meaningfully improve the clarity, utility, and positioning of our paper.

---

### Decision · Action_Editor_pvSQ · 2025-07-02

**Recommendation:** Accept as is

**Additional Comments:**

This paper presents a wide-ranging overview of generative AI, offering practitioners a clear understanding of current capabilities and limitations, and I support the reviewers' consensus on acceptance—provided all requested revisions have been thoroughly addressed in the final version.

**Audience:**

Yes

**Audience Explanation:**

The topic of the paper is well suited for a segment of the TMLR audience.

**Claims And Evidence:**

Yes

**Claims Explanation:**

The paper positions itself as a survey on generative AI, and the points it makes are clearly explained and supported.